# The Oxford Needle Experience (ONE) scale: a UK-based and US-based online mixed-methods psychometric development and validation study of an instrument to assess needle fear, attitudes and expectations in the general public

Jonathan Kantor [1,2,3] Samantha Vanderslott,[1] Michael Morrison,[4] Andrew J Pollard [1] Robert C Carlisle[2]

For numbered affiliations see end of article.

**Correspondence to**
Jonathan Kantor;
jonkantor@gmail.com

## ABSTRACT

**Objectives** To develop and validate the Oxford Needle Experience (ONE) scale, an instrument to assess needle fear, attitudes and expectations in the general population.

**Design** Cross-sectional validation study.

**Setting** Internet-based with participants in the UK and USA.

**Participants** UK and US representative samples stratified by age, sex, and ethnicity using the Prolific Academic platform.

**Main outcome measures** Exploratory factor analysis with categorical variables and a polychoric correlation matrix followed by promax oblique rotation on the UK sample for the ONE scale. Confirmatory factor analysis (CFA) with a Satorra-Bentler scaled test statistic evaluating the root mean squared error of approximation (RMSEA), standardised root mean squared residual (SRMR) and comparative fit index (CFI) on the US sample. Reliability as internal consistency using McDonald's omega. Convergent validity using the Pearson correlation coefficient. Predictive and discriminant validity using logistic regression ORs of association (OR).

**Results** The population included 1000 respondents, 500 in the UK and 500 in the USA. Minimum average partial correlation and a scree plot suggested four factors should be retained: injection hesitancy, blood-related hesitancy, recalled negative experiences and perceived benefits, yielding a 19-question scale. On CFA, the RMSEA was 0.070 (90% CI, 0.064 to 0.077), SRMR 0.053 and CFI 0.925. McDonald's omega was 0.92 and 0.93 in the UK and US samples, respectively. Convergent validity with the four-item Oxford Coronavirus Explanations, Attitudes and Narratives Survey (OCEANS) needle fear scale demonstrated a strong correlation (r=0.83). Predictive validity with a single-question COVID-19 vaccination status question demonstrated a strong association, OR (95% CI) 0.97 (0.96 to 0.98), p<0.0001 in the US sample. Discriminant validity with a question regarding the importance of controlling what enters the body confirmed

## STRENGTHS AND LIMITATIONS OF THIS STUDY

⇒ Using an iterative qualitative methodological approach to domain development permits the inclusion of a broad and diverse stakeholder group in the development of the Oxford Needle Experience scale.

⇒ Relying on two large age-, sex- and racially representative populations from both the UK and USA may improve the generalisability of our findings.

⇒ Adopting an approach that relies on polychoric correlation matrices while treating Likert-scale responses as ordinal variables may mitigate some of the statistical risks associated with treating ordinal data as continuous, thereby providing more reliable and valid results in our analysis of complex behavioural and attitudinal patterns.

⇒ The reliance on survey panel design may introduce selection bias, as despite being demographically representative the sample may not accurately represent the general population's attitudes and experiences.

⇒ The study focuses on the UK and US populations, potentially limiting the scale's generalisability to other cultural contexts, and additional cross-cultural validation studies may be warranted.

the ONE score does not predict this unrelated outcome, OR 1.00 (0.99, 1.01), p=0.996 in the US sample.

**Conclusions** The ONE scale is a reliable and valid multidimensional scale that may be useful in predicting vaccine hesitancy, designing public health interventions to improve vaccine uptake and exploring alternatives to needles for medical procedures.

## INTRODUCTION

Vaccine hesitancy was identified by the WHO as one of the top 10 threats to public health

even before the onset of the COVID-19 pandemic, and the potential impact of this phenomenon on global health has only increased in recent years.[1] While vaccine hesitancy is a complex and multifactorial phenomenon, previous work has suggested that injection fear may represent its only modifiable risk factor.[2 3] Given the importance of behavioural considerations in modulating vaccine acceptance, changes in methods of vaccine delivery may decrease vaccine hesitancy, leading to significant global health benefits.[4] With upwards of 10 billion vaccinations given by needle each year, better understanding the role of needles as potential contributors to vaccine hesitancy is of urgent public health importance, particularly as alternative approaches for vaccine delivery are undergoing development.[5–7]

Despite the central importance of understanding public attitudes to needle use for vaccination and other medical procedures, little is known regarding the myriad contributors to this phenomenon, and most previous scales designed to study needle attitudes, experiences and fear were developed without fundamental qualitative research, in the absence of a multidimensional approach—often focusing exclusively on needle fear, anxiety or even phobia—or in populations with suboptimal generalisability.[8–15]

Validated multidimensional scales are a useful tool to study behavioural phenomena, and may provide insight with direct global health implications.[16] Our goal, therefore, was to develop a feasible multidimensional scale that would capture the key domains contributing to concern regarding needle use in the context of vaccination or medical procedures. By better elucidating the underlying domains that may modulate vaccine hesitancy, and medical procedure hesitancy more broadly, these data may ultimately be used to better inform the development of evidence-based vaccination strategies to mitigate some of the perceived negative aspects of vaccine delivery. Moreover, a better understanding of the contribution of needles to vaccine hesitancy may also encourage the development of alternative vaccine-delivery technologies, ultimately resulting in more tailored and publicly acceptable approaches to vaccination and medical procedures in the future.

## METHODS
### Domain and item generation
Domain generation to establish the domains of interest for the ONE scale was accomplished through an iterative deductive and inductive approach: after a literature review was performed, online focus groups were used to qualitatively develop a set of possible themes, and ultimately domains, that putatively contribute to needle attitudes, experiences and expectations. Online focus groups were conducted iteratively using two separate 50-respondent samples recruited through Prolific Academic. The first was a convenience sample in the UK, and the second included a group of respondents who

had previously answered that they had not received any vaccinations for COVID-19. The subject was introduced by stating that 'The aim of this study is to create a survey that can be used to understand attitudes towards needles and vaccine administration via needles. The reason we are asking you to complete the survey is that in order to develop valid surveys for future use it is important for researchers to determine whether the questions that we are asking are reasonable, relatable, and understandable. It is also important for us to look at the ways in which asking similar questions in slightly ways may affect responses, and how the answers to these questions group together. This will enable us to develop meaningful survey instruments in the future that can be used to better assess public attitudes'. Open-ended questions included items such as 'What are five words that come to mind when you think about needles?' In addition, respondents were presented with multiple choice questions and asked to provide feedback on question quality, clarity, and saliency.

To assess the face and content validity of the scale, extensive qualitative measures were performed. The two 50-member online focus groups were asked, 'Did you feel like this set of multiple-choice questions was easy to understand? Reasonable?' to gain insights into the clarity, relevance and interpretability of the items. This feedback mechanism was instrumental in refining the items to ensure they are aligned with the intended constructs and are easily understood. Furthermore, the iterative workshopping of items among the investigators, the Oxford Vaccine Group, the full BUBBL group at the Department of Engineering Science and biology/ in vivo cluster meetings enriched the scale's content validity. Each item was meticulously reviewed and refined based on the collective feedback, ensuring a comprehensive and representative measure of individuals' perceptions and experiences regarding needles and injections.

A group of experts from the fields of medicine, public health, social science and engineering science was consulted regarding domain development and inclusion, consistent with established scale development methodology.[16–18]

Participant item responses were recorded on a 5-point Likert scale ranging from strongly agree to strongly disagree. Items were reverse scored, as appropriate, to yield a scale where higher values represent greater levels of concern regarding needle use. This work was approved by the University of Oxford Medical Sciences Interdivisional Research Ethics Committee (approval reference R81585/RE001). All subjects provided informed consent for participation and were permitted to withdraw from their anonymous surveys at any time. Qualitative analyses were performed using NVivo release V.1.4 (QSR International, Burlington, Massachusetts, USA) and statistical analyses were performed using Stata for Mac V.16 (Stata Corporation, College Station, Texas, USA).

## Data collection

Two separate representative samples were used in the development of the ONE. A sample of 500 respondents in the UK was recruited through Prolific Academic (Oxford, UK), an academic survey panel site and a draft scale was presented using the Qualtrics platform (Qualtrics Corp, Provo, Utah, USA). All respondents were rewarded with a small payment (<£2). This sample was used to develop the scale and perform exploratory factor analysis (EFA). A second demographically representative sample of respondents from the USA was then solicited in a similar fashion using the same electronic survey recruitment tools. This sample was used for confirmatory factor analysis and reliability and validity testing. Results from the USA and UK were evaluated separately to establish the robustness of the scale. Demographic information was self-reported by respondents and was linked with responses through an anonymous 25-digit alphanumeric key. Baseline information included age, self-reported sex, self-reported race and ethnicity, employment status, education history, household income and vaccination status against COVID-19.

## Domain and item reduction and exploratory factor analysis

The Kaiser-Meyer-Olkin procedure was conducted to ensure sampling adequacy.[19] To establish the number of domains to retain for the ONE scale, we calculated the minimum average partial correlation for the number of principal components, developed a scree plot and performed parallel analysis for principal components.[18 20 21] Item reduction was approached by assessing domain relevance, examining polychoric (item-item) and polyserial (item-test) correlations, culling redundant items and removing those with low communality or low factor loadings.[22] EFA with categorical variables was performed by using polychoric correlations to generate a factor matrix using iterated principal factor analysis followed by promax oblique factor rotation.[23 24]

## Confirmatory factor analysis

The robustness of the findings was tested by performing confirmatory maximum likelihood factor analysis using structural equation modelling on the US sample with the model developed in the UK sample with a Satorra-Bentler scaled test statistic. Multiple goodness of fit indices were analysed, including the root mean squared error of approximation (RMSEA), the standardised root mean squared residual (SRMR) and the comparative fit index (CFI) to assess model fit. Values of RMSEA$\leq$0.08, SRMR$\leq$0.08 and CFI$\geq$0.90 suggest acceptable fit.[25 26]

## Reliability and validity

Reliability as internal consistency was assessed by examining McDonald's omega, a more robust alternative to Cronbach's alpha, for each subscale domain and for the overall scale, in both the UK and US samples.[27]

Convergent validity was evaluated by examining the correlation between the ONE scale and the four-item OCEAN fears factor score.[3] Predictive validity was assessed by examining the association between the ONE scale score and vaccination status against COVID-19. Logistic regression ORs of association were calculated with vaccination status (defined as having received a minimum of one dose of any vaccine for COVID-19) coded as a dichotomous variable and the ONE scale treated as a continuous predictor variable. Discriminant validity was assessed by examining the logistic regression ORs of association between the ONE score and agreement with a non-specific statement ('it is important to control what enters one's body') in the UK and US samples. A p-value of 0.05 was considered significant.

## Patient and public involvement

This study was conducted to develop a scale to assess public attitudes to needle use for medical procedures such as vaccination. Patient and public involvement was not actively solicited for the initial phase of study design, and patients and members of the public were primarily participating as survey respondents. However, patients and members of the public taking part in online focus groups on needle hesitancy were able to provide feedback on the aims of the project as well as the suitability of the wording of specific scale items. The aim of this research is to develop a reliable scale that reflects the priorities and preferences of the public. Results will be disseminated and may potentially be used in future studies to better assess patient and public preferences regarding the use of needles in vaccination and in medical procedure more broadly.

## RESULTS

### Respondent characteristics

In the UK sample, of the 506 respondents who started the survey, 500 subjects returned completed surveys, yielding a 98.8% completion rate in the survey panel. The median (IQR) age was 45 (32, 59), and respondents ranged in age from 18 to 90; 50.4% of respondents identified as female, 48.2% as male and 1% as non-binary. The median (IQR) self-reported household income was £39 000 (25 000 to 58 000). A total of 47.6% of respondents were currently working full-time and 14.2% identified as non-white.

In the US sample, of the 509 respondents who started the survey, 500 subjects returned completed surveys, yielding a 98.2% completion rate in the survey panel. The median (IQR) age was 45 (31, 60), and respondents ranged in age from 18 to 84; 49.6% of respondents identified as female, 47.6% as male and 1.6% as non-binary. The median (IQR) self-reported household income was $53 500 (31 000 to 89 999). A total of 45.5% of respondents were currently working full-time and 27.8% identified as non-white.

### Domain and item reduction and exploratory factor analysis

A total of 16 themes were identified as potentially contributing to public attitudes to needles after literature review and qualitative analysis of online focus group data. The

**Table 1** Factor loadings for the ONE EFA

| Domain | Item | Factor 1 | Factor 2 | Factor 3 | Factor 4 |
|---|---|---|---|---|---|
| Injection | I am afraid of needles. | **0.9449** | 0.0468 | −0.0622 | −0.0951 |
| | I am afraid of being vaccinated because of the needle. | **0.9808** | −0.1095 | −0.0307 | 0.0119 |
| | I have experienced severe panic when thinking about or undergoing an injection. | **0.7896** | 0.0227 | 0.1669 | 0.0064 |
| | When I know I will need to be injected with a needle, I worry about it ahead of time. | **0.9088** | −0.0146 | 0.0406 | −0.0914 |
| | I have delayed getting a vaccine specifically because of the needle. | **0.7954** | −0.0051 | 0.0474 | 0.1659 |
| | I am bothered by the appearance of a needle. | **0.7300** | 0.2277 | −0.0493 | −0.0410 |
| | I feel anxious at the thought of being injected with a needle. | **0.8713** | 0.1421 | −0.0257 | −0.0505 |
| | I am worried that I will feel unwell or faint if I am injected with a needle. | **0.5640** | 0.2972 | 0.0736 | −0.0564 |
| | Needles cause significant pain. | **0.6236** | −0.1182 | 0.0984 | 0.1585 |
| | I dislike that a needle pierces my skin and enters my body directly. | **0.7196** | −0.0200 | 0.0426 | 0.0897 |
| Blood | I am afraid of seeing blood. | 0.2112 | **0.6760** | −0.0008 | 0.0197 |
| | I avoid watching surgical procedures on television or on the internet. | 0.1448 | **0.5628** | −0.0812 | −0.0174 |
| | I feel lightheaded when I see blood. | −0.0415 | **0.9135** | 0.0460 | 0.0736 |
| History | I had a bad experience with needles in the past. | 0.0861 | 0.0572 | **0.8677** | 0.0616 |
| | I have a strong unpleasant memory of being injected with a needle. | 0.3179 | −0.0119 | **0.7238** | −0.0835 |
| | I remember a nurse or doctor making a mistake when injecting me with a needle. | −0.1582 | 0.0021 | **0.8672** | −0.0221 |
| Benefits | I think needles are useful. | 0.1967 | −0.0099 | −0.0248 | **0.5859** |
| | My feelings regarding needles do not matter since needles are so useful. | −0.0694 | 0.0645 | 0.0150 | **0.7389** |
| | People should try to overcome their fear of needles because of the benefits of using them. | −0.0999 | 0.0274 | −0.0146 | **0.8757** |

Values greater than 0.35 are in bold.
EFA, exploratory factor analysis; ONE, Oxford Needle Experience.

| Table 2 | Factor intercorrelations in the UK EFA | | | |
|---|---|---|---|---|
| | **Injection** | **Blood** | **History** | **Benefits** |
| Injection | 1 | | | |
| Blood | 0.59 | 1 | | |
| History | 0.50 | 0.20 | 1 | |
| Benefits | 0.36 | 0.20 | 0.15 | 1 |
| EFA, exploratory factor analysis. | | | | |

themes identified through a literature review included fear, history of negative experiences, intrusive thoughts, risk, avoidance, needle appearance, fainting, anxiety, pain, blood and trust. The focus group feedback added several additional themes including general dislike, concerns regarding personal control, fears regarding maintaining bodily integrity, positive aspects of needle use and the training/expertise/skill of the injector. A total of 107 items for potential scale inclusion were then developed for testing; the intentionally large domain and item pool was designed to pilot similar iterations of items in a focus group format. A total of 88 items were ultimately deleted due to low domain relevance, low polychoric or polyserial correlations, unifactorial outcomes, redundancy, or low communality.

Factor loadings from the 19 retained items after EFA for categorical variables with promax oblique rotation are presented in table 1. The Kaiser-Meyer-Olkin measure of sampling adequacy was excellent, with a value of 0.92 for the overall model and with values ranging from 0.68 to 0.97 for each of the individual items. This four-factor solution reflected a simple structure with appropriate loading and was supported by the results of the minimum average partial correlation analysis, the scree plot and the Eigenvalue >1 approaches to factor extraction.[28 29] Parallel analysis suggested retaining three factors, though this result was considered borderline given the adjusted Eigenvalue of 0.97 for the fourth factor, just below the cut-off of 1.[20]

Factors correlated clinically with the domains of (1) injection hesitancy (including fear and anxiety related to injection); (2) blood-related hesitancy; (3) recalled negative experiences and (4) perceived benefits of needles.

Factor intercorrelations are included in table 2, and the final ONE scale and scoring key are presented in table 3.

### Confirmatory factor analysis
Confirmatory factor analysis was performed on the US sample using the model developed in the UK sample. The maximum likelihood factor analysis model created using structural equation modelling demonstrated acceptable goodness of fit characteristics, with RMSEA=0.070 (90%

| Table 3 | The ONE scale | |
|---|---|---|
| **Number** | **Item** | **Domain** |
| 1 | I am afraid of needles. | Injection |
| 2 | I am afraid of being vaccinated because of the needle. | Injection |
| 3 | I have experienced severe panic when thinking about or undergoing an injection. | Injection |
| 4 | When I know I will need to be injected with a needle, I worry about it ahead of time. | Injection |
| 5 | I have delayed getting a vaccine specifically because of the needle. | Injection |
| 6 | I am bothered by the appearance of a needle. | Injection |
| 7 | I feel anxious at the thought of being injected with a needle. | Injection |
| 8 | I am worried that I will feel unwell or faint if I am injected with a needle. | Injection |
| 9 | Needles cause significant pain. | Injection |
| 10 | I dislike that a needle pierces my skin and enters my body directly. | Injection |
| 11 | I am afraid of seeing blood. | Blood |
| 12 | I avoid watching surgical procedures on television or on the internet. | Blood |
| 13 | I feel lightheaded when I see blood. | Blood |
| 14 | I had a bad experience with needles in the past. | History |
| 15 | I have a strong unpleasant memory of being injected with a needle. | History |
| 16 | I remember a nurse or doctor making a mistake when injecting me with a needle. | History |
| 17 | I think needles are useful. | Benefits |
| 18 | My feelings regarding needles do not matter since needles are so useful. | Benefits |
| 19 | People should try to overcome their fear of needles because of the benefits of using them. | Benefits |
| All answer choices are rated using a 5-point Likert scale (strongly disagree though strongly agree). For questions 1-16, strongly agree is scored as 5, while for questions 17-19 strongly agree is scored as 1 (reverse scoring). Higher values reflect more hesitancy regarding needle use. The total possible ONE score ranges from 19 to 95. ONE, Oxford Needle Experience. | | |

**Table 4** Median and IQR for ONE subscales and the overall scale

| | UK sample Median (IQR) | US sample Median (IQR) | P value* |
|---|---|---|---|
| Injection | 17 (13–26) | 19 (13–29) | 0.07 |
| Blood | 7 (4–10) | 7 (4–9) | 0.80 |
| History | 6 (4–10) | 8 (4–12) | <0.001 |
| Benefits | 5 (4–6) | 5 (4–7) | 0.32 |
| ONE scale overall | 37 (29–48) | 39 (29–54) | 0.054 |

*P-values are for the two-sample Wilcoxon rank-sum (Mann Whitney) test.
ONE, Oxford Needle Experience.

CI 0.064 to 0.077), SRMR=0.053 and CFI=0.925 using the four-factor model with a total of 19 items. Standardised factor loadings are presented in the online supplemental table.

The median (IQR) total ONE score was 37 (29–48) in the UK population and 39 (29–54) in the US population (p=0.054). Further descriptive statistics for each subscale are included in table 4.

### Reliability and validity

Reliability as internal consistency was excellent: McDonald's omega for the overall ONE scale was 0.92 in the UK sample and 0.93 in the US sample. Omega values for each subscale are included in table 5.

Convergent validity with the OCEAN needle fear score supported our validity argument, showing a strong correlation between the scales (r=0.83). Predictive validity with the single-question vaccination status question demonstrated a significant association between vaccination status and ONE score, with a logistic regression OR of association (95% CI) of 0.98 (0.96 to 1.0), p=0.014 for the UK sample and 0.97 (0.96 to 0.98), p<0.0001 for the US population. Given that the SD of the ONE score in the UK and US samples is 15.2 and 16.5, respectively, this translates into a decrease in the odds of being vaccinated of 30.4% in the UK and 49.5% in the USA for each single SD increase in ONE score.

Discriminant validity with a question regarding the importance of controlling what enters the body confirmed that the ONE score does not predict this unrelated outcome, with logistic regression ORs of association

**Table 5** Subscale and overall reliability, as measured by McDonald's Omega

| Subscale | UK sample | US sample |
|---|---|---|
| Injection | 0.94 | 0.93 |
| Blood | 0.79 | 0.79 |
| History | 0.85 | 0.84 |
| Benefits | 0.71 | 0.70 |
| Overall | 0.92 | 0.93 |

of 1.01 (1.00, 1.02), p=0.162 in the UK population and 1.00 (0.99, 1.01), p=0.996 in the US sample.

### DISCUSSION

We have shown here that the ONE scale is a reliable multidimensional instrument and that there is significant preliminary evidence to support the scale's validity for assessing public attitudes to needle use in the context of medical procedures such as vaccination. Given that injection fears may be the primary modifiable risk factor for vaccine refusal,[3] understanding the contributors to such responses is of vital public health importance. Moreover, needle concerns are not limited to those undergoing vaccination, and have been implicated in hesitancy regarding a range of procedures from venepuncture[30] to local anaesthetic injection in dermatologic surgery,[31] blood donation,[32] dental procedures[33] and dialysis.[34] Therefore, a better appreciation of the myriad contributors to the needle exposure experience may be broadly valuable.

In evaluating descriptive statistics for each of the subscales, only the median scores for a recalled history of negative experiences differed significantly between the UK and US populations, with an absolute 33% increase reported in the US group (p<0.0001). As such recall has been previously implicated as a potential driver of needle fear—where a feedback loop of recalled negative experiences leads to avoidance, furthering fear—this may have important implications for needle acceptance.[5] Moreover, the magnitude of the association between ONE score and COVID-19 vaccination status—the odds of being unvaccinated increase by almost 50% for each SD increase in ONE score in the US population—both highlights the predictive validity of the scale and also underscores its utility and potential value in designing public health interventions and the potential role of needle hesitancy as a contributor to vaccine hesitancy.[3]

Several prior instruments have been developed to assess needle attitudes. The 18-item Injection Phobia Scale-Anxiety (IPS-Anx) was developed in 1992 on a sample of 40 subjects with known needle phobia and examines fear and anxiety exclusively.[15] Two decades after its initial development, the factorial structure of the scale was assessed, suggesting a two-factor structure, encompassing contact (direct injection) and distal (watching or discussing injection) fears.[35] The 50-item Medical Fear Survey examines an array of fears relating to medical procedures, including injections as well as blood, physical examination and mutilation;[36] an abbreviated version of this survey has also been developed.[10] The blood/injection fear scale was developed on a Turkish population of patients attending a hospital for examination or blood donation and is a 20-question scale that includes two domains, injection and blood.[12] The Blood-Injection Symptom Scale assesses domains of faintness, anxiety and tension, and relied on a population of Australian university students for its development.[8] The 40-item Multidimensional Blood/

Injury Phobia Inventory (MBPI) was developed on a US university student population to assess phobias, and is aimed at assessing pathological reactions, rather than understanding the range of experiences of the general population exposed to needles.[9 11] Blood-injection-injury (BII) was also assessed as part of a 10-item subscale of the Specific Phobia Questionnaire, a lengthy scale addressing phobias ranging from fear of animals to the natural environment; this scale was developed on a cohort of patients with known anxiety disorders and a group of Canadian university students.[13] Finally, the multidimensional fear of injection scale was developed on 419 Japanese university students to evaluate fear of injections exclusively, encompassing domains of direct fear, indirect fear, physiological response, and avoidance.[14]

Unlike these scales, we used a qualitative methodology to ensure that our domain-generation and item-generation would include the diversity of priorities of a wide range of respondents to the experience of undergoing injection with a needle—a methodology designed to capture the broadest possible themes that may not be obvious to investigators a priori.[17 37–39] This is important, as prior studies have suggested a disconnect between clinicians' perception of patient or public priorities and their actual priorities.[40 41] The ONE scale is also the only instrument to include domains that assess recalled negative experiences as well as positive beliefs regarding the utility of needles, which may add significantly to researchers' understanding of patient and public priorities.

Additional strengths of our study include the iterative approach and the use of representative populations in our exploratory and confirmatory factor analysis; many prior studies relied on less diverse populations of young, well-educated students—or, conversely, on those being treated for anxiety disorders—and therefore the generalisability of their scales to the wider population is unknown. Furthermore, some prior scales aimed to evaluate pathological responses to needle injection— needle phobia. While such work is highly valuable, our goal was to develop a scale that could be broadly adopted for designing public health interventions and alternative vaccine and medical therapeutic delivery technologies for the public.

We also relied on statistical analyses appropriate for ordinal data, including a polychoric-matrix based EFA, to provide conservative estimates given concerns regarding the assumed normality of Likert-scale data.[42 43] This represents a departure from earlier studies that acknowledged the skewness inherent in needle fear assessments— the majority of individuals have only mild symptoms, while a small proportion have extreme reactions—but used statistics more appropriate for analysing normally distributed data with continuous endpoints.[11]

The samples from the UK and USA were designed to be demographically representative. Prolific Academic creates representative samples in the UK and US populations by stratifying by age (across five 9 year age brackets), sex (male or female) and ethnicity (using the five simplified categories recommended by the UK Office of National Statistics: White, Mixed, Asian, Black and Other). Cross stratifying by age (five brackets), sex (two groups) and ethnicity (five groups) yields 50 subgroups. The number of respondents in each subgroup is designed to match census data from the US Census Bureau (2015) or the UK Office of National Statistics (2011), as appropriate, and allocations to each subgroup are made proportionately. Subjects are recruited from the survey panel site, whose current database includes over 130 000 identity-confirmed participants, and subjects are offered to participate if the particular subgroup to which they belong (eg, an Asian male between 18 and 27) has not yet been filled. We acknowledge that a panel-based sampling method has biases, though the large sample size and broad range of respondent characteristics suggests that this approach yields insights that are broadly applicable to the general populations of the UK and USA.

Our work has several important limitations. First, we relied on a survey panel design to allow for a demographically representative population from both the UK and USA, but there may be important differences between the general population and the subset willing to volunteer to contribute to survey panels, and response rates cannot be calculated with this approach. Moreover, all research that relies on self-report introduces the risk that bias, and particularly response bias and social desirability bias, may affect stated preferences. Nevertheless, the robustness of our findings, with total and subscale scores largely similar across both the UK and US populations, supports the generalisability of the scale. Second, one domain—injection hesitancy—accounts for 10 of the 19 items in the scale, while the remaining domains each include three items. The unequal number of items across domains, however, was purposely designed to capture the presence of a primary domain of interest—injection hesitancy—while also acknowledging and evaluating the contribution of other domains. Indeed, some have suggested that a unitary construct underlies responses to needle injection,[11] and it was our goal to expand on this central domain rather than attempt to over-extract factors that otherwise correlate closely. Third, additional validation testing, as well as test–retest reliability and cross-cultural validation studies, should be considered in the future to better understand the limits of the ONE scale's generalisability, particularly given the potential for cultural and spiritual beliefs to affect these results. Fourth, an additional limitation of our manuscript is the absence of invariance testing and multigroup comparisons. We did not assess whether the ONE scale's item scores are comparable between subgroups based on demographic variables, such as country of origin, age and socioeconomic status, or determine whether the scale operates equivalently across different groups regarding the perceived item content and the dimensionality of underlying constructs. Future research including such analyses may aid in establishing the scale's robustness across diverse clinical and research settings. Finally, an

added limitation of our methodological approach was the lack of formal cognitive interviews performed to establish face and content validity. While we acknowledge that our approach does not provide an absolute guarantee of item interpretation consistency, the thorough and iterative review and feedback process bolsters the face and content validity of the ONE scale. Though the focus group data and feedback from experts was instrumental in developing the scale, future studies including cognitive interviews may be helpful to further support arguments for the validity of the ONE scale for assessing needle attitudes.

The ONE scale addresses four separate domains that contribute to the public's experience with needles: injection hesitancy, blood-related hesitancy, recalled negative experiences and perception of the benefits associated with needle use. Distinguishing between these domains and appreciating their independence and interdependence may provide helpful insights in designing public health interventions. Indeed, as with any multidimensional scale, when interpreting the total ONE score, it is important to appreciate that different domains (subscales) contribute in varying amounts to the scoring and may offset each other. Injection hesitancy accounts for the lion's share of the variance, and includes 10 of the 19 total score item, but this can be offset by scores on the other domains. While the correlations between the injection subscale and blood-related hesitancy subscale are fairly strong (0.59), some of the other subscales correlate poorly—for example, the correlation between the perceived benefits of needle use and other subscales ranges from 0.15 to 0.36. On a practical level, this means that while subscale data add important granular detail, they may also substantially offset each other, potentially diluting the actionable utility of the global score in understanding contributors to needle attitudes. Therefore, as with any multidimensional scale, the overall score provides useful summary information but for deeper insights attention must be directed at the subscale scoring level. The ONE scale represents a new parsimonious instrument that may provide important actionable insights to promote global public health, and further research and testing may be warranted.

All answer choices are rated using a 5-point Likert scale (strongly disagree though strongly agree). For questions 1–16, strongly agree is scored as 5, while for questions 17–19 strongly agree is scored as 1 (reverse scoring). Higher values reflect more hesitancy regarding needle use. The total possible ONE score ranges from 19 to 95.

**Author affiliations**
[1]Oxford Vaccine Group, University of Oxford, Oxford, UK
[2]Biomedical Ultrasonics, Biotherapy, and Biopharmaceuticals Laboratory (BUBBL), Institute of Biomedical Engineering, Department of Engineering Science, University of Oxford, Oxford, UK
[3]Department of Dermatology, Center for Global Health, and Center for Clinical Epidemiology and Biostatistics, University of Pennsylvania Perelman School of Medicine, Philadelphia, Pennsylvania, USA
[4]Centre for Health, Law, and Emerging Technologies (HeLEX), University of Oxford, Oxford, UK

**Contributors** Study conception: JK and RCC. Study design: JK, SV, MM, AJP and RCC. Statistical analyses: JK, AJP and RCC. Writing: JK, SV, MM, AJP and RCC. Oversight: SV, MM, AJP and RCC. Guarantor: JK.

**Funding** SV received funding from AHRC-IRC (HNR02360) and NIHR (HNR02910). RCC is grateful for the generous benefaction of Mr Donald Porteus which supported his contribution. The funding agencies and parties were not involved in study design, data collection, analysis or presentation.

**Competing interests** None declared.

**Patient and public involvement** Patients and/or the public were involved in the design, or conduct, or reporting or dissemination plans of this research. Refer to the Methods section for further details.

**Patient consent for publication** Not required.

**Ethics approval** This study involves human participants and was approved by University of Oxford Medical Sciences Interdivisional Research Ethics Committee (approval reference R81585/RE001). Participants gave informed consent to participate in the study before taking part.

**Provenance and peer review** Not commissioned; externally peer reviewed.

**Data availability statement** Data are available upon reasonable request. Data are available from the corresponding author.

**ORCID iDs**
Jonathan Kantor http://orcid.org/0000-0002-3256-3014
Andrew J Pollard http://orcid.org/0000-0001-7361-719X

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
