## [Reviewer comments · BMJ Open]

ARTICLE DETAILS

TITLE (PROVISIONAL)	The Oxford Needle Experience (ONE) scale: a UK- and US-based online mixed-methods psychometric development and validation study of an instrument to assess needle fear, attitudes, and expectations in the general public
AUTHORS	Kantor, Jonathan; Vanderslott, Samantha; Morrison, Michael; Pollard, Andrew; Carlisle, Robert C

VERSION 1 – REVIEW

REVIEWER	Jaracz, Krystyna Poznan University of Medical Sciences
REVIEW RETURNED	17-Jul-2023

GENERAL COMMENTS	Thank you to the Editor for inviting me to review this manuscript. The paper is interesting and needed from both theoretical and practical perspectives, providing a valuable contribution to existing knowledge by offering academics and healthcare professionals a new tool to assess concerns and attitudes towards needle injections for vaccine or medication administration. In the introduction and discussion, the authors present convincing arguments justifying the need to develop an additional measurement scale regarding the use of needles for medical purposes and procedures. A significant strength of the study presented in the reviewed manuscript is the application of a mixed approach, combining quantitative and qualitative methods to develop the substantive content of the scale. It is also noteworthy that the ONE scale does not solely focus on negative emotions and fears related to needle pricks but allows for a deeper exploration of people's concerns associated with injections. The research methods, including data collection of both quantitative and qualitative data, as well as the statistical analyses, were appropriately chosen and clearly and comprehensively described in the manuscript. The use of non-parametric statistical analyses in EFA, statistical descriptions, and comparative analyses is particularly worth highlighting. Another positive aspect of the reviewed study is the use of representative populations, enabling the authors to generalise the findings to a broader general population. The obtained results, presented concisely yet clearly and comprehensively in both the text and tables, provide convincing evidence of the sound psychometric properties of the ONE scale. In my opinion, the final conclusions drawn from the study are fully justified.
--

	In summary, this paper presents a needed and interesting study. I have no questions or critical comments. I thoroughly enjoyed reviewing the manuscript. Congratulations to the authors, and I will recommend its publication.
--	--

REVIEWER	Poot, Charlotte Leiden University Medical Center, Department of Public Health and Primary Care
REVIEW RETURNED	06-Oct-2023

GENERAL COMMENTS	This a well written paper describing the development and validity testing of a self-report scale to measure needle concerns (ONE scale). The strength of the questionnaire lies 1) in the holistic approach towards needle fear and anxiety, resulting in a multidimensional scale that is widely applicable and 2) the large representative sample which is used for both development and validity testing. The authors use robust and standard practice study methods and statistical analysis approaches, however the paper mainly focuses on psychometric analysis. I recommend the authors to address the following points: Major concerns:  1) Page 8, line 39 – line 40: The authors report on the number of themes that were identified after literature review and focus group discussion (FGD) with a number of experts. However it is unclear what these items were and which items were derived from literature review (as known theme's) and which were added after the FGD analysis. Please, clarify to improve transparency of the process. 2) Page 8, line 44: Regarding the FGD more information should be given here. How was it conducted? Which questions were asked? How was the subject introduced beforehand? What was there input? Details should be added to the method and result section. 3) Page 6: Items were formulated however not tested on face or content validity using for example cognitive interviewing. How can we be certain that the items are interpreted as intended (i.e. underlying cognitive process) or that the likert-scale is an appropriate scale to respond to the items ? Please comment on the face and content validity of the ONE scale and how you have assessed this. 4) Page 9: Results are presented on internal structure validity using CFA. Please also report on the (standardized) factor loadings of the individual items to aid interpretation and support future cross-cultural validation 5) Age and social-economical position are known to influence needle fear/concerns. The authors conclude that the ONE scale is widely applicable. However the authors have not assessed whether item scores are comparable between subgroups (based on demographic variables of interest) or whether the scales operates equivalent across groups with respect to perceived item content and dimensionality of the underlying constructs/dimension (i.e. internal structure, factorial structure). I advise the authors to perform a multi-group comparison using invariance testing. Please refer also to the paper by B.M. Byrne (Testing across nations and cultures: issues and Complexities, 2015). Given that the dataset is a good representation of future respondents of the ONE scale I believe the dataset is suitable for invariance testing. If the authors decide on not performing the additional analysis, the authors should include the absence of invariance testing and multi-group comparison as an important limitation of the study and comment on the implications for use in practice. 6) The authors present validity as a characteristic of the instrument. However, current leading belief is that tests, or instruments are
---

	themselves not valid or invalid, but rather are valid for a particular use. Hence, we encourage the authors to consider this validity assessment study as initial evidence on the validity of the ONE scale. Which, however, which requires future evidence from multiple source so strengthen the 'validity argument'. (see The standards for Educational and Psychological Testing). 7) Page 7, line 50: Convergent validity was tested by examining the association between the ONE scale score and vaccination status against COVID-19. By default, convergent testing should be performed to test how closely related the test is to another test to measure the same (or similar construct). In the introduction the authors refer to the paper by Freeman et al, 2023 on injection fears and COVID-19, 2023. However, the authors of this study concluded that only 11,5 % of all instance of vaccine hesitancy was attributable to blood-injection-phobia. Following the line or reasoning concerning the 'measuring of similar constructs' and the observed AF, I question whether similar theoretical constructs are measured. I advise to comment on the theorized similarity of the tests and the justification for convergent testing. 8) Needle fear and anxiety is predominantly discussed in light of vaccination and vaccination hesitancy. The authors identify 'blood-related hesitancy' as a separate domain and construct within in there four-factor structure. Please comment on how needle fear for injections (such as vaccines) and needle fear for blood-drawing relates/ differs and what its implications are for use (e.g. can people score high on domain injection-hesitancy and low on blood-related hesitancy) . Please comment on its implication for total score interpretation and consider domain scoring in practical use of the questionnaire. Also, in some cultures blood-drawing hesitancy is strongly correlated with cultural and spiritual beliefs. Minor comments: 1) Page 10, line 12: Comparable instruments are presented. To strengthen the justification of a new and better instrument it is stronger to present the argument straight after the discussion of the instrument instead of first presenting a whole list of instruments.
--	---

VERSION 1 – AUTHOR RESPONSE

Reviewer: 1

Dr. Krystyna Jaracz, Poznan Univ Med Sci

Comments to the Author:

Thank you to the Editor for inviting me to review this manuscript.

The paper is interesting and needed from both theoretical and practical perspectives, providing a valuable contribution to existing knowledge by offering academics and healthcare professionals a new tool to assess concerns and attitudes towards needle injections for vaccine or medication administration.

In the introduction and discussion, the authors present convincing arguments justifying the need to develop an additional measurement scale regarding the use of needles for medical purposes and procedures.

A significant strength of the study presented in the reviewed manuscript is the application of a mixed approach, combining quantitative and qualitative methods to develop the substantive content of the scale. It is also noteworthy that the ONE scale does not solely focus on negative emotions and fears related to needle pricks but allows for a deeper exploration of people's concerns associated with injections.

The research methods, including data collection of both quantitative and qualitative data, as well as the statistical analyses, were appropriately chosen and clearly and comprehensively described in the manuscript. The use of non-parametric statistical analyses in EFA, statistical descriptions, and comparative analyses is particularly worth highlighting.

Another positive aspect of the reviewed study is the use of representative populations, enabling the authors to generalise the findings to a broader general population.

The obtained results, presented concisely yet clearly and comprehensively in both the text and tables, provide convincing evidence of the sound psychometric properties of the ONE scale. In my opinion, the final conclusions drawn from the study are fully justified.

In summary, this paper presents a needed and interesting study. I have no questions or critical comments. I thoroughly enjoyed reviewing the manuscript. Congratulations to the authors, and I will recommend its publication.

We thank the reviewer for their positive comments. We very much appreciate that the reviewer shares our belief in the importance of approaching scale development in a rigorous fashion, and we thank them again for their enthusiastic reception of our manuscript.

Reviewer: 2

Dr. Charlotte Poot, Leiden University Medical Center

Comments to the Author:

This a well written paper describing the development and validity testing of a self-report scale to measure needle concerns (ONE scale). The strength of the questionnaire lies 1) in the holistic approach towards needle fear and anxiety, resulting in a multidimensional scale that is widely applicable and 2) the large representative sample which is used for both development and validity testing. The authors use robust and standard practice study methods and statistical analysis approaches, however the paper mainly focuses on psychometric analysis. I recommend the authors to address the following points:

Major concerns:

1) Page 8, line 39 – line 40: The authors report on the number of themes that were identified after literature review and focus group discussion (FGD) with a number of experts. However it is unclear what these items were and which items were derived from literature review (as known theme's) and which where added after the FGD analysis. Please, clarify to improve transparency of the process.

We thank the reviewer for their insightful and sophisticated question.

We have amended the manuscript by adding the following:

The themes identified through a literature review included fear, history of negative experiences, intrusive thoughts, risk, avoidance, needle appearance, fainting, anxiety, pain, blood, and trust. The focus group feedback added several additional themes including general dislike, concerns regarding personal control, fears regarding maintaining bodily integrity, positive aspects of needle use, and the training/expertise/skill of the injector.

2) Page 8, line 44: Regarding the FGD more information should be given here. How was it conducted? Which questions were asked? How was the subject introduced beforehand? What was there input? Details should be added to the method and result section.

We thank the reviewer for their close reading. We have added this detail to the body of the manuscript as follows:

Online focus groups were conducted iteratively using two separate 50-respondent samples recruited through Prolific Academic. The first was a convenience sample in the UK, and the second included a group of respondents who had previously answered that they had not received any vaccinations for COVID-19. The subject was introduced by stating that 'The aim of this study is to create a survey that can be used to understand attitudes towards needles and vaccine administration via needles. The reason we are asking you to complete the survey is that in order to develop valid surveys for future use it is important for researchers to determine whether the questions that we are asking are reasonable, relatable, and understandable. It is also important for us to look at the ways in which asking similar questions in slightly ways may affect responses, and how the answers to these questions group together. This will enable us to develop meaningful survey instruments in the future that can be used to better assess public attitudes.' Open ended questions included items such as 'What are five words that come to mind when you think about needles?' In addition, respondents were presented with multiple choice questions and asked to provide feedback on question quality, clarity, and saliency.

3) Page 6: Items were formulated however not tested on face or content validity using for example cognitive interviewing. How can we be certain that the items are interpreted as intended (i.e. underlying

cognitive process) or that the likert-scale is an appropriate scale to respond to the items ? Please comment on the face and content validity of the ONE scale and how you have assessed this.

Thank you for another excellent question. While we did not perform cognitive interviewing, we did include items in the focus group rounds and specifically ask the following question after each set of multiple-choice questions: 'Did you feel like this set of multiple-choice questions was easy to understand? Reasonable? Please provide any comments regarding the question quality, and if you think that something else should be included so that we can better understand people's feelings about needles and being injected with needles.' Answers to open-ended questions suggested that respondents sought response options that both gauged their degree of (dis)agreement while also desiring a neutral option, suggesting that a 5-point Likert scale was appropriate. Additionally, items were workshopped not only among the investigators, but were also presented to larger groups for feedback, including the Oxford Vaccine Group and the full BUBBL group at the Department of Engineering Science as well as the biology/ in vivo cluster meetings in Engineering Science. Specific feedback regarding item structure and interpretability was sought iteratively. That said, this approach does not guarantee that items were interpreted as intended, and we have added this to the limitations section.

To further clarify the face and content validity of the ONE scale, we have added the following paragraph to the manuscript:

To assess the face and content validity of the scale, extensive qualitative measures were performed. The two 50-member online focus groups were asked, 'Did you feel like this set of multiple-choice questions was easy to understand? Reasonable?' to gain insights into the clarity, relevance, and interpretability of the items. This feedback mechanism was instrumental in refining the items to ensure they are aligned with the intended constructs and are easily understood. Furthermore, the iterative workshopping of items among the investigators, the Oxford Vaccine Group, the full BUBBL group at the Department of Engineering Science, and biology/ in vivo cluster meetings enriched the scale's content validity. Each item was meticulously reviewed and refined based on the collective feedback, ensuring a comprehensive and representative measure of individuals' perceptions and experiences regarding needles and injections.

Added to the limitations section:

Finally, an added limitation of our methodological approach was the lack of formal cognitive interviews performed to establish face and content validity. While we acknowledge that our approach does not provide an absolute guarantee of item interpretation consistency, the thorough and iterative review and feedback process bolsters the face and content validity of the ONE scale. Though the focus group data and feedback from experts was instrumental in developing the scale, future studies including cognitive interviews may be helpful to further support arguments for the validity of the ONE scale for assessing needle attitudes.

4) Page 9: Results are presented on internal structure validity using CFA. Please also report on the (standardized) factor loadings of the individual items to aid interpretation and support future cross-cultural validation.

Thank you for raising this important point. We have now included the standardized factor loadings as an additional table for reference as Table 4. Tables 4 and 5 in the original manuscript are now listed as Tables 5 and 6.

Latent Variable	Item	Standardized Factor Loadings	Std. Err.	z-value	P-value
	Injection 1	1 (constrained)	- - -		
	Injection 2	0.897	0.043	20.77	<0.0001
	Injection 3	0.878	0.044	19.92	<0.0001
	Injection 4	1.149	0.042	27.43	<0.0001
	Injection 5	0.673	0.052	12.99	<0.0001
	Injection 6	0.997	0.042	23.78	<0.0001
	Injection 7	0.859	0.052	16.50	<0.0001
	Injection 8	1.180	0.043	27.52	<0.0001
	Injection 9	0.658	0.048	13.68	<0.0001
	Injection 10	0.955	0.048	19.73	<0.0001
	Blood 11	1 (constrained)	- - -		
	Blood 12	0.928	0.063	14.80	<0.0001
	Blood 13	1.008	0.057	17.58	<0.0001
	History 14	1 (constrained)	- - -		
	History 15	1.135	0.044	25.89	<0.0001
	History 16	0.837	0.051	16.36	<0.0001
	Benefits 17	1 (constrained)	- - -		

Benefits 18 1.860 0.243 7.66 <0.0001

Benefits 19 1.850 0.236 7.83 <0.0001

Table 4. Standardized factor loadings of the ONE scale using structural equation modeling.

5) Age and social-economical position are known to influence needle fear/concerns. The authors conclude that the ONE scale is widely applicable. However the authors have not assessed whether item scores are comparable between subgroups (based on demographic variables of interest) or whether the scales operates equivalent across groups with respect to perceived item content and dimensionality of the underlying constructs/dimension (i.e. internal structure, factorial structure). I advise the authors to perform a multi-group comparison using invariance testing. Please refer also to the paper by B.M. Byrne (Testing across nations and cultures: issues and Complexities, 2015). Given that the dataset is a good representation of future respondents of the ONE scale I believe the dataset is suitable for invariance testing. If the authors decide on not performing the additional analysis, the authors should include the absence of invariance testing and multi-group comparison as an important limitation of the study and comment on the implications for use in practice.

Thank you for raising the important issue of invariance testing and multi-group comparison. We appreciate that this is an important limitation of the manuscript, and to maintain the 'readability' of the manuscript for the non-specialist clinician audience of BMJ Open, as highlighted in the instructions for authors, rather than including these analyses we instead – as you suggest – have now included their absence as an important limitation of our study. The limitations section has been amended as below:

Fourth, an additional limitation of our manuscript is the absence of invariance testing and multi-group comparisons. We did not assess whether the ONE scale's item scores are comparable between subgroups based on demographic variables, such as country of origin, age, and socioeconomic status, or determine whether the scale operates equivalently across different groups regarding the perceived item content and the dimensionality of underlying constructs. Future research including such analyses may aid in establishing the scale's robustness across diverse clinical and research settings.

6) The authors present validity as a characteristic of the instrument. However, current leading belief is that tests, or instruments are themselves not valid or invalid, but rather are valid for a particular use. Hence, we encourage the authors to consider this validity assessment study as initial evidence on the validity of the ONE scale. Which, however, which requires future evidence from multiple source so strengthen the 'validity argument'. (see The standards for Educational and Psychological Testing).

Thank you for highlighting this important and subtle methodological point. We note the elegant way in which the reviewer has previously addressed this issue, for example in their own work regarding the Dutch version of the eHealth literacy questionnaire. In order to highlight that validity is not, as noted in the Standards, an inherent characteristic of a scale, but rather something for which evidence is being provided for (and against), we have amended our discussion so that references to the scale as being valid as a blanket statement have been deleted or amended in favour of discussing the validity argument for or against its use.

For example, in the discussion section, a sentence now reads : We have shown here that the ONE scale is a reliable multidimensional instrument and that there is significant preliminary evidence to support the scale's validity for assessing public attitudes to needle use in the context of medical procedures such as vaccination.

7) Page 7, line 50: Convergent validity was tested by examining the association between the ONE scale score and vaccination status against COVID-19. By default, convergent testing should be performed to test how closely related the test is to another test to measure the same (or similar construct). In the introduction the authors refer to the paper by Freeman et al, 2023 on injection fears and COVID-19, 2023. However, the authors of this study concluded that only 11,5 % of all instance of vaccine hesitancy was attributable to blood-injection-phobia. Following the line or reasoning concerning the 'measuring of similar constructs' and the observed AF, I question whether similar theoretical constructs are measured. I advise to comment on the theorized similarity of the tests and the justification for convergent testing.

Thank you for your insightful comment, and we appreciate the opportunity to clarify the type of validity we are demonstrating. The reviewer is absolutely correct: the data presented supported the predictive validity, rather than convergent validity, of the ONE scale. While we feel that including the predictive validity further bolsters our validity argument (as alluded to above), we have corrected our wording to highlight that those data support predictive validity and now include data on the convergent validity of the scale as well.

We have therefore amended the manuscript as follows to explicitly discuss convergent and predictive validity:

In the methods:

Convergent validity was evaluated by examining the correlation between the ONE scale and the 4-item OCEAN fears factor score.³

In the results:

Convergent validity with the OCEAN needle fear score supported our validity argument, showing a strong correlation between the scales ($r=0.83$).

Our intention was to demonstrate that the ONE scale has practical utility in predicting real-world behaviors associated with needle fears, such as vaccination hesitancy. We believe that establishing this link helps add to the validity argument, given that one of the primary applications of the scale may be to identify individuals at risk of avoiding vaccination due to needle fear and develop targeted interventions for this group, even though—as noted by the reviewer—needle fear is not the primary driver of vaccine hesitancy.

8) Needle fear and anxiety is predominantly discussed in light of vaccination and vaccination hesitancy. The authors identify 'blood-related hesitancy' as a separate domain and construct within in there four-factor structure. Please comment on how needle fear for injections (such as vaccines) and needle fear for blood-drawing relates/ differs and what its implications are for use (e.g. can people score high on domain injection-hesitancy and low on blood-related hesitancy) . Please comment on its implication for total score interpretation and consider domain scoring in practical use of the questionnaire. Also, in some cultures blood-drawing hesitancy is strongly correlated with cultural and spiritual beliefs.

We thank the reviewer for yet another excellent comment. Indeed, while the injection subscale accounts for the greatest proportion of the variance (and has the largest number of items) in the scale overall, the other domains do indeed contribute separately to the overall score. We have now commented on the implications for the multidimensional nature of the scale and practical implications by adding the following to the discussion:

Indeed, as with any multidimensional scale, when interpreting the total ONE score it is important to appreciate that different domains (subscales) contribute in varying amounts to the scoring and may offset each other. Injection hesitancy accounts for the lion's share of the variance, and includes 10 of the 19 total score item, but this can be offset by scores on the other domains. While the correlations between the injection subscale and blood-related hesitancy subscale is fairly strong (0.59), some of the other subscales correlate poorly—for example, the correlation between the perceived benefits of needle use and other subscales ranges from 0.15 to 0.36. On a practical level, this means that while subscale data add important granular detail, they may also substantially offset each other, potentially diluting the actionable utility of the global score in understanding contributors to needle attitudes. Therefore, as with any multidimensional scale, the overall score provides useful summary information but for deeper insights attention must be directed at the subscale scoring level.

Finally, we appreciate the reviewer's comment regarding cultural and spiritual beliefs around blood-related hesitancy. We have amended the following sentence in our limitations to address this issue:

Third, additional validation testing, as well as test-retest reliability and cross-cultural validation studies, should be considered in the future to better understand the limits of the ONE scale's generalisability, particularly given the potential for cultural and spiritual beliefs to affect these results.

Minor comments:

1) Page 10, line 12: Comparable instruments are presented. To strengthen the justification of a new and better instrument it is stronger to present the argument straight after the discussion of the instrument instead of first presenting a whole list of instruments.

We thank the reviewer for an additional thoughtful comment. While we initially considered commenting on the strengths and weaknesses of each of the prior instruments and discussing the relative merits of the ONE scale compared to each instrument individually, given word limit constraints and the editors' concerns regarding manuscript readability (as well as the fact that several of the prior instruments shared similar shortcomings), we felt it would be more straightforward to present the list of prior scales and then enter continue the discussion of the merits of the ONE scale in this context. If the reviewer or editors feel strongly we are, of course, more than happy to amend the structure of our discussion.

REVIEWER	Poot, Charlotte Leiden University Medical Center, Department of Public Health and Primary Care
REVIEW RETURNED	10-Nov-2023

GENERAL COMMENTS	After a thorough re-review, I am pleased to inform you that I have no further comments to add. The authors have addressed all my previous concerns, and the revisions have significantly strengthened the overall quality of the manuscript. I believe that this paper will make a valuable contribution to BMJ Open, and I recommend it for publication without reservation. The authors have demonstrated a high level of expertise in their field, and their research adds meaningful insights to the existing body of knowledge. Once again, I would like to commend the authors for their dedication to improving the manuscript and for their thoughtful responses to my previous feedback.
--